Promoting effect of rapamycin on osteogenic differentiation of maxillary sinus membrane stem cells

Lin Yanjun 1 3
Zhang Min 2 4
Zhou Lin 3
Chen Xuxi 3
Chen Jiang 3
Wu Dong wudong@fjmu.edu.cn 3
1 Fujian Key Laboratory of Oral Diseases, Fujian Medical University , Fuzhou , Fujian , China
2 Fujian Provincial Engineering Research Center of Oral Biomaterial, Fujian Medical University , Fuzhou , Fujian , China
3 Research Center of Dental and Craniofacial Implants, Fujian Medical University , Fuzhou , Fujian , China
4 General Department of Hubin Campus, Hangzhou Stomatology Hospital , Hangzhou , Zhejiang , China
Uversky Vladimir
Electronic publication date: 2021 Jun 1
Publication date: 2021
Volume: 9
Electronic Location ID: e11513
Received 2020 Jun 22; Accepted 2021 May 4
Copyright: ©2021 Lin et al.
Copyright year: 2021
Copyright holder: Lin et al.
License: This is an open access article distributed under the terms of the Creative Commons Attribution License, which permits unrestricted use, distribution, reproduction and adaptation in any medium and for any purpose provided that it is properly attributed. For attribution, the original author(s), title, publication source (PeerJ) and either DOI or URL of the article must be cited.
License URL: https://creativecommons.org/licenses/by/4.0/

Keywords: Maxillary sinus membrane stem cells, Rapamycin, Osteogenic differentiation, Autophagy

Funding: Natural Science Foundation of Fujian Province 2018J01819 The work was supported by the Natural Science Foundation of Fujian Province (2018J01819). The funders had no role in study design, data collection and analysis, decision to publish, or preparation of the manuscript.

==============================
Background

Stem cells located in the maxillary sinus membrane can differentiate into osteocytes. Our study aimed to evaluate the effect of rapamycin (RAPA) on the osteogenic differentiation of maxillary sinus membrane stem cells (MSMSCs).

Methods

Colony-forming unit assay, immunophenotype identification assay, and multi-differentiation assay confirmed characteristics of MSMSCs obtained from SD rats. Transmission electron microscopy (TEM) and flow cytometry (FCM) identified the initial autophagic level of MSMSCs induced by RAPA. Real-time quantitative PCR (qPCR) evaluated subsequent autophagic levels and osteogenic differentiation. Alkaline phosphatase (ALP) activity assay and alizarin red staining (ARS) evaluated subsequent osteogenic differentiation. We performed a histological examination to clarify in vivo osteogenesis with ectopic bone mass from BALB/c nude mice.

Results

MSMSCs possessed an active proliferation and multi-differentiation capacity, showing a phenotype of mesenchymal stem cells. The autophagic level increased with increasing RAPA (0, 10, 100, 1,000 nM) and decreased over time. ALP activity and calcium nodules forming in four RAPA-treated groups on three-time points (7, 14, 21 d) showed significant differences. Col1a1, Runx2, and Spp1 expressed most in 100 nM RAPA group on 7 and 14 d. Osteogenesis-related genes except for Ibsp expression between four groups tended to be consistent on 21 d. 100 nM and 10 nM RAPA-treated groups showed more bone formation in vivo.

Conclusion

RAPA can promote osteogenic differentiation of MSMSCs, indicating a possible relationship between osteogenic differentiation and autophagy.

Introduction

Maxillary sinus membrane elevation is a frequent option when bone height in the posterior maxilla is less than four mm. Maxillary sinus membrane elevation consists of two situations, bone augmentation with or without bone substitutes. Our previous research observed satisfying alveolar bone gains and a nearly 100% survival rate of implants using a maxillary sinus membrane elevation technique (Chen et al., 2016). When we elevate the maxillary sinus membrane without bone substitutes, there is still new bone formation underneath the membrane, even though it is less than that when elevating with bone substitutes (Yan et al., 2018).

For the time being, there are some interpretations of new bone formation underneath the Schneiderian membrane after surgery. Bone marrow stromal cells (BMSCs) immigrating from jaw marrow stroma to its surface can produce new bone, which may be the primary cell source (Chen et al., 2018). Meanwhile, blood with productive growth factors can provide scaffolds and activators (Bonazza et al., 2018). Additionally, maxillary sinus membrane stem cells (MSMSCs) feature in the formation of new bone underneath the maxillary sinus membrane (Berbéri et al., 2017; Srouji et al., 2010). The maxillary sinus membrane is different from the periosteum. It consists of four layers (pseudostratified ciliated columnar epithelium, basement membrane, lamina propria, and periosteum-like structure) (Srouji et al., 2009). MSMSCs from the periosteum-like layer can differentiate into an osteogenic lineage and express osteogenic markers such as ALP, RUNX2, OCN, ON, and BSP (Guo et al., 2015). MSMSCs, similar to other mesenchymal stem cells (MSCs), are a reliable source of bone formation. Regardless of some studies on MSMSCs, examples of applications are rare, and the possible mechanisms of osteogenic differentiation remain unknown. In a previous study, MSCs showed a high level of basal autophagy that decreased during differentiation (Oliver et al., 2012). An additional study indicated that the maintenance of cellular stemness and the lineage determination of MSCs were deeply affected by autophagy (Sbrana et al., 2016). Rapamycin (RAPA) is a specific mTOR inhibitor that activates autophagy and promotes osteogenic differentiation of BMSCs (Wan et al., 2017).

Several papers reported that autophagy drove osteogenic differentiation of mesenchymal stem cells. (Pantovic et al., 2013; Vidoni et al., 2019; Wan et al., 2017). Up to date, no study has focused on the relationship between autophagy and osteogenic differentiation of MSMSCs. Moreover, no study has focused on the effect of RAPA on MSMSCs. Therefore, the effect of RAPA on osteogenic differentiation of MSMSCs is an issue to investigate.

This study aimed to delve into the effect of RAPA on the osteogenic differentiation of MSMSCs. Colony-forming unit (CFU) assay, multi- differentiation assay, and immunophenotype identification assay evaluated cellular stemness. The autophagic level was validated by transmission electron microscopy (TEM) and flow cytometry (FCM). Real-time quantitative PCR (RT-qPCR) monitored autophagic markers (Becn1, LC3). Bone formation ex vivo was evaluated by alkaline phosphatase (ALP) activity assay and alizarin red staining (ARS). Furthermore, RT-qPCR monitored osteogenic markers (Col1a1, Ibsp, Runx2, Spp1). Ectopic bone formation assay evaluated bone formation in vivo.

Materials & Methods

Animals

A total of SD male rats (N = 50) and BALB/c male nude mice (N = 20) (SLAC Laboratory Animal Co., Ltd., Shanghai, China) were housed in the Department of Comparative Medicine, the 900th Hospital of Joint Logistics Support Force of the Chinese People’s Liberation Army. No more than five animals per cage were in pathogen-free and sterilized space with Sani-chip bedding and a standard light-dark cycle. The environment temperature was 23 ± 2 °C, and the humidity was 50–60% with proper ventilation. According to rodent standard feeding rules, the animals were fed with standard laboratory tap water and commercial purified diets. SD rats were sacrificed using ten times the dose of conventional anesthesia with pentobarbital (1% sodium pentobarbital, 40 mg/kg ) by intraperitoneal injection to obtain maxillary sinus membrane cells. We established an ectopic subcutaneous osteogenesis model using BALB/c nude mice. BALB/c nude mice were euthanized with pentobarbital (1% sodium pentobarbital, 50 mg/kg) by intraperitoneal injection after one month of feeding to obtain the specimen. To induce loss of consciousness and death with a minimum of pain and distress, they were euthanized with overdose pentobarbital (10 times of general anesthesia) after surgery. Rigor mortis confirmed the death. After obtaining the specimen, the animal carcasses were collectively processed by the animal center. Criteria established for euthanizing animals before the planned end of the experiment followed AVMA guidelines. All animal procedures followed the NIH guide of Humane Use and Care Animals and were approved by the Institutional Ethics Committee of the School of Stomatology, Fujian Medical University (Ref. [2017] NO.59).

Cell culture

Maxillary sinus membrane samples were separated carefully from SD rats. The tissues were cut into pieces and digested in a solution composed of 3 mg/mL collagenase type I and 4 mg/mL dispase (Sigma, USA) for 50 min at 37 °C. Samples were pooled and filtered by a 70 µm strainer (Falcon, USA) to obtain single-cell suspensions. Single-cell suspensions (1 × 104 cells) were seeded into 75 cm2 culture dishes (Corning, USA) with alpha modification of Eagle’s medium (HyClone, USA) supplemented with 15% fetal bovine serum(GIBCO, USA), 100 U/ml penicillin, and 100 mg/ml streptomycin (HyClone, USA). Suspensions were incubated at 37 °C in 5% CO2. Once cells reached 80% confluence, they were passaged with a ratio of 1:3. The cells at passage three were used in the following assays. Before osteogenic differentiation induction, cells were pretreated with RAPA (0, 10, 100, 1,000 nM) for four hours.

CFU assay

1 × 103 MSMSCs at passage 3 were cultured for seven days in 60 cm2 dishes and stained with 0.1% crystal violet. Aggregates of ≥50 cells were scored as colonies to assess colony-forming efficiency.

Multi-differentiation assay

Differentiation media kits (Cyagen, China) were used to induce osteogenic, adipogenic, and chondrogenic differentiation. Respectively, cells at densities of 1 × 104 cells/cm2 (osteogenic induction), 3. 2 × 104 cells/cm2 (adipogenic induction) or 5 × 105 cells/cm2 (chondrogenic induction) were seeded. Cells were fixed in 4% paraformaldehyde for 15 min after 28 d (osteogenic induction), 16 d (adipogenic induction), and 21 d (chondrogenic induction). Osteocytes were stained for 5 min with an alizarin red solution (pH 4.2). Adipocytes were stained for 15 min with oil red O. Chondrocytes were stained for 15 min with alcian blue.

Immunophenotype identification assay

A total of 1 × 106 MSMSCs at passage 3 were harvested and suspended in 500 µl of PBS containing 20 ng/ml AlexaFluor® 488-coupled antibodies against CD34, CD45, CD90, CD146, and CD166. After incubation for 30 min at 4 °C, the cells were washed twice in PBS and analyzed by flow cytometry using the BD Accuri C6.

Autophagy TEM analysis

We treated the passage 3 MSMSCs seeded on glass slides with different RAPA (0, 10, 100, and 1,000 nM) for four hours. Cells were fixed in 2.5% glutaraldehyde for 2 h at 4 °C and washed in phosphate-buffered saline (PBS). Cells were then fixed with 1% K2[OsO2(OH)4]⋅0.1 M PBS (pH 7.4) for 2 h. Sections (70 nm) were imaged on a transmission electron microscope (Tecnai G2 F20 S-TWIN, FEI, USA), and we captured autophagosomes as described in the results.

Autophagy FCM analysis

We treated the passage 3 MSMSCs with four groups of RAPA (0, 10, 100, and 1,000 nM) for four hours. 1 × 106 cells were suspended in 500 µl of PBS containing 50 nM monodansylcadaverine (MDC). After incubation for 30 min at 4 °C, the cells were washed with PBS and resuspended in 1 ml of PBS for analysis using the BD Accuri C6.

ALP activity assay

The passage 3 MSMSCs at densities of 2 × 104 cells/well were seeded into 96-well plates (Corning, USA) and treated with four groups of RAPA (0, 10, 100, and 1,000 nM) for four hours. According to the manufacturer’s instructions, after 7, 14, and 21 days of culture in the calcification medium, ALP activity was detected using an ALP assay kit (Beyotime, China). The amount of ALP in the cells was normalized against the total protein content.

Quantitative ARS assay

The passage 3 MSMSCs at densities of 1 × 104 cells/cm2 were seeded into 6-well plates (Corning, USA) and treated with four groups of RAPA (0, 10, 100, and 1,000 nM) for four hours. After 7, 14, and 21 days of culture in calcification medium, cells were fixed in 4% paraformaldehyde for 15 min and de-stained in 10 mmol L−1 sodium phosphate containing 10% cetylpyridinium chloride (Sigma, USA). The amount of alizarin red was quantified by the absorbance of the solution at 562 nm using a microplate reader.

RT-qPCR

According to the manufacturer’s instructions, total RNA was isolated from passage 3 MSMSCs using Trizol reagent (TaKaRa, Japan). The purity and concentration of RNA were identified by UV spectroscopy. The Genomic DNA was excluded using gDNA Eraser in PrimeScript™ RT reagent Kit (TaKaRa, Japan) at 42 °C for 2 min, then cDNA was synthesized with 1 µg of total RNA. Real-time polymerase chain reaction assays were performed on triplicate samples using SYBR Premix Ex Taq™ II (Takara, Japan) in a Roche 480 Light Cycler (Roche, Mannheim, Germany). The cycling conditions consisted of incubating at 95 °C for 30 s, 40 cycles of 95 °C for 5 s, and 60 °C for 30 s. Gapdh as an internal control, we calculated relative expression levels of indicated genes by the 2−ΔΔCt method. Primer sequences were as follows (Table 1):

Table 1 Primer sequences for real-time RT-PCR.

Gene	Access	Forward	Reverse	
Gapdh	NM_017008	GTATGACAATGAATATGGCTACAG	TCTCTTGCTCTCAGTATCCTTG	
LC3	NM_012823	GGCTCTGGCTATTCTGTCTC	CTGACTTACATCTGGTGCTGAA	
Becn1	NM_001034117	GCGGCTCCTATTCCATCAA	AGCATCTTTCCAAACCAAACAAA	
Col1a1	NM_053304	AAAGATGGACTCAACGGTCTC	CAGGAAGCTGAAGTCATAACCA	
Ibsp	NM_012587	AACCTTAGCCGTTCAGATGT	CAGACACCACTGTAACCTAGAA	
Runx2	NM_001278483	GGACCGACACAGCCATAT	GGAAGGATGAGAGCCAACT	
Spp1	NM_012881	CAGACACCACTGTAACCTAGAA	TTGCCTGCCTCTACATACATT	

Ectopic bone formation assay

MSMSCs at passage 3 pretreated with different concentrations of RAPA were harvested and mixed with bone substitute materials (Bio-Oss®, Geistlich, Switzerland). We transplanted the cell pellets into the flanks of nude mice. Four weeks after implantation, we terminated the mice, and harvested scaffolds for H&E, Masson, and Goldner trichrome staining. Briefly, we cut the paraffin blocks into four mm sections; for H&E staining, the sections were stained with Harris’ hematoxylin and eosin for 2 min; for Masson staining, the sections were treated with a Masson staining fluid (Solarbio, China) for 5 min; for Goldner staining, the sections were treated with a Goldner staining fluid (Solarbio, China) for 5 min. We observed the results under a microscope.

Statistical analysis

Each experiment was performed in triplicate and repeated at least three times. We compared experimental groups using a two-way analysis of variance or repeated-measures ANOVA with SPSS 22.0 software. Values were presented as means ± SD. All p-values were two-tailed, and p < 0.05 was considered statistically significant.

Results

Characterization of MSMSCs

Primary cells were successfully isolated from the SD rat maxillary sinus mucosa. They showed a spindle-shaped and fibroblast-like morphology, similar to BMSCs. MSMSCs possessed the active ability of proliferation and reached confluence after three days of culture in complete medium. 1 ×103 single cells were seeded at low density, forming 49 ± 4.0 colonies (1CFU-F per 204 cells) (Fig. 1A). Trilineage differentiation experiments validated the multi-differentiation potential of MSMSCs, including osteogenesis, adipogenesis, and chondrogenesis. After osteogenic induction for 28 days, we observed calcium granules stained by Alizarin Red in MSMSCs. After adipogenic induction for 14 days, we observed lipid droplets stained by Oil Red O in MSMSCs. After chondrogenic induction for 14 days, we observed a sulfated proteoglycan-rich matrix stained by Alcian blue in MSMSCs (Fig. 1B). Flow cytometry identified the phenotype of MSMSCs. MSMSCs positively expressed MSC markers, CD90 and CD146, while negatively expressing hematopoietic cell markers, CD34, and CD45. Otherwise, MSMSCs of SD rats negatively expressed CD105 and CD166 (Fig. 1C).

Figure 1 Characterization of MSMSCs.

(A) Representative image of a single colony-forming unit of MSMSCs. Scale bars represent 200 µm and 80 µm, respectively. (B) Representative images of the osteogenic, adipogenic, and chondrogenic differentiation of MSMSCs. Scale bars represent 100 µm. (C) Cell surface markers of MSMSCs; representative figures of cytometric flow tests and percentage of positive expression.

Validation of autophagy in MSMSCs

Different concentrations of autophagy activator (0 nM, 10 nM, 100 nM, and 1,000 nM RAPA) were used to pretreat MSMSCs for 4 h prior to osteogenic differentiation. We observed autophagosomes by TEM (Fig. 2A). As the concentration increased, the number of autophagosomes in the cells gradually increased. Notably, in this study, MSMSCs treated with 1,000 nM RAPA showed the highest autophagosome concentrations. Interestingly, we found many amorphous organelles, but unexpectedly, these were at least partially encapsulated in autophagosomes. Additionally, the autophagosomes did not seem to be full of lysosomes. FCM detected autophagic vacuoles stained by MDC (Fig. 2B). Cells with 0 nM RAPA were left untreated to probe for basal autophagosome recycling. The green fluorescence intensity of MSMSCs represented the autophagic level in the cell. The autophagic stimulated ratios were 1.33% of the 0 nM RAPA-treated cells, 43.3% of the 10 nM RAPA-treated, 78.9% of the 100 nM RAPA-treated and 98.0% of 1,000 nM RAPA-treated cells. The control group possessed a relatively low basal level of autophagy. Decuple RAPA concentration change was capable of notably altering the autophagic state of MSMSCs. Becn1 and LC3 expression represented subsequent levels of autophagy (Fig. 2C). The autophagic level after 7d of osteogenic induction was still dose-dependent. The autophagic level was the highest in the 1,000 nM RAPA-treated group. However, over time in the osteogenic culture, the difference in Becn1 and LC3 expression between the four groups became less.

Figure 2 Autophagic levels of MSMSCs after pretreatment with four concentrations of RAPA for 4 h.

(A) Representative figures of transmission electron microscopy. Scale bars represent 1 µm. (B) Representative figures of cytometric flow tests and percentage of positive expression. (C) The gene expression levels of the autophagic marker LC3 were examined by qPCR. (D) The gene expression levels of the autophagic markers Becn1 were examined by qPCR. Relative mRNA expression was normalized to the control. The values are expressed as the mean ±  SD. *p < 0.05.

RAPA promoted ex vivo osteogenic differentiation of MSMSCs

MSMSCs were induced to osteogenic differentiation for 7, 14, and 21 days after pretreatment with RAPA for 4 h. BCIP/NBT staining represented the dynamic state of ALP. The outcomes showed that ALP expression in the four concentration groups was, in descending order, 100 nM RAPA, 1,000 nM RAPA, 10 nM RAPA, and 0 nM RAPA. However, the difference between each group gradually decreased, developing with further differentiation. On the 21st day of osteogenic differentiation, the visible differences between the four groups were insignificant. ALP activity is an essential indicator of osteoblast differentiation. Therefore, we performed a quantitative ALP activity assay (Fig. 3A). ARS assessed calcium deposits. Calcium deposits increased and then reduced after pretreatment with RAPA (Fig. 3B). These changes were also dose-dependent. Real-time quantitative PCR was employed to detect the mRNA expression of Col1a1, Ibsp, Runx2, and Spp1 (Fig. 3C). The expression levels of osteogenesis-related genes changed with RAPA treatment. This finding indicated that RAPA could alter the development of osteogenic differentiation. In the early stage of osteogenic differentiation, MSMSCs mainly expressed Runx2, and the Runx2 expression levels of the four groups gradually achieved the same level, which indicated that the effect of RAPA on osteogenesis gradually decreased. The change in expression of Col1a1 was the same as Runx2 expression. Regarding the expression of Spp1 in the whole osteogenic stage, the 100 nM RAPA groups had the highest Spp1 expression, and the 10 nM RAPA and 1,000 nM RAPA groups had similar expression levels. The expression of Ibsp, in the late stage of osteogenic differentiation, showed a difference similar to that of OPN expression.

Figure 3 RAPA upregulated ex vivo osteogenic differentiation in MSMSCs.

(A) Representative images of four groups over three-time points using BCIP/NBT staining. Scale bars represent 100 µm. (B) Representative images of the four groups over three-time points using Alizarin Red staining. Scale bars represent 100 µm. (C) ALP activities were detected by the alkaline phosphatase assay kit. (D) The total areas of mineralized nodules in the different groups were quantified using cetylpyridinium chloride. (E) The gene expression levels of the osteogenic marker Col1a1 were examined by qPCR. (F) The gene expression levels of the osteogenic marker Ibsp were examined by qPCR. (G) The gene expression levels of the osteogenic marker Runx2 were examined by qPCR. (H) The gene expression levels of the osteogenic marker Spp1 were examined by qPCR. Relative mRNA expression was normalized to the control. All the values above are expressed as the mean ± SD. *p < 0.05.

RAPA promoted in vivo ectopic bone formation of MSMSCs

To further examine the effect of autophagy on ectopic bone formation in vivo, we transplanted immunocompromised mice with MSMSCs pretreated with four concentrations of RAPA. Four weeks after transplantation, the bony masses that formed at the surgical sites were removed and analyzed. H & E (Fig. 4A), Masson’s trichrome (Fig. 4B), and Goldner’s trichrome staining (Fig. 4C) revealed that RAPA might induce bone-like tissue formation in vivo. Similar to native bone tissue, the new bone-like structure contained osteocytes encased in the newly deposited bone matrix. We observed little bone formation in the RAPA-free groups. Interestingly, the numbers of new vessels formed in the 100 nM and the 10 nM RAPA-treated groups seemed more than that in the other two groups. It indicated that an appropriate RAPA treatment might have improved the new bone formation and improved the formation of new blood vessels, which created a virtuous circle of local bone formation.

Figure 4 RAPA increased in vivo ectopic bone formation in MSMSCs.

Nude mice were implanted with MSMSCs treated with four concentrations of RAPA integrated with Bio-Oss®. New bone-like structures (NB) with osteocyte-like cells (arrow) embedded within the calcified matrix was formed on the surface of Bio-Oss® carrier. Connective tissue (CT) surrounded the carriers. (A) Histologic evaluation of H&E-stained paraffin-embedded tissue sections. (B) Histologic evaluation of Masson’s Trichrome-stained paraffin-embedded tissue sections. (C) Histologic evaluation of Goldner’s Trichrome-stained paraffin-embedded tissue sections. Scale bars represent 50 µm.

Discussion

Osteogenic differentiation of MSMSCs was confirmed many years ago, indicating that the Schneiderian membrane possesses osteogenic potential (Guo et al., 2015). In this study, MSMSCs expressed MSCs relevant immunocytochemical markers and possessed the ability of osteogenic differentiation. Evidence shows that MSMSCs do not express CD105 and CD166 but do positively express CD90 and CD146. Reduced expression of CD90 can increase the level of osteogenic differentiation of MSCs. CD146 can be used to identify subpopulations of cells with osteogenic capacity in MSCs. We divided the osteogenic differentiation of MSCs into four stages: immature osteoprogenitor cells, mature osteoprogenitor cells, pre-osteoblasts, and mature bone cells. In these four stages, multiple factors regulate MSCs, and Runx2 is a crucial factor. Knocking out Runx2 in animal models results in the absence of skeletal systems in experimental mice (Takarada et al., 2013). RAPA can induce transcriptional activation of osteogenic differentiation through increased GATA4 and Sox17 that modulate downstream Runx2 (Gambacurta et al., 2019). Also, some markers often express in the osteogenic differentiation of MSCs. For example, ALP does not express in immature osteoprogenitor cells, but ALP expression increases gradually with osteogenic differentiation (Srouji et al., 2010). Col1a1 is not expressed merely in immature osteoprogenitor cells (Rasi et al., 2016). Ibsp expresses in all four stages, but the difference in expression is vast (Bouleftour et al., 2016). The specific performance of osteogenic differentiation is the enhancement of Alp, Col1a1, Ibsp, Spp1, and Runx2 gene expression. Therefore, in this study, MSMSCs were subjected to osteogenic induction after RAPA pretreatment, and then Col1a1, Ibsp, Runx2, and Spp1 were selected as the target genes. A variety of factors are involved in regulating autophagosome formation, and LC3 is a crucial marker of autophagy (Tanida, Ueno & Kominami, 2008). Besides, BECN1 is also one of the critical proteins regulating autophagy and can form complexes with related proteins to induce autophagy and regulate autophagy levels (Kang et al., 2011). Autophagy refers to the formation of autophagosomes by forming a membrane structure that encapsulates part of the cytoplasm, intracellular bodies, and proteins that need to degrade due to damage and senescence. Autophagy lysosomes achieve the physiological processes of cell homeostasis and organelle renewal by degrading their encapsulation (Mizushima & Komatsu, 2011). This process also plays a role in the fate decision of MSCs (Green & Levine, 2014). Osteogenic differentiation is a prevailing differentiation direction of MSCs, and studies have shown that autophagy plays a regulatory role in this process (Pantovic et al., 2013). The occurrence of autophagy is affected by many factors, and cells can increase the level of autophagy following induction by factors such as hypoxia, stress, and drug treatment (Ravanan, Srikumar & Talwar, 2017). In the ex vivo experiments modeling autophagy, drug induction is the most common method for changing autophagy. Among them, autophagy agonists such as RAPA (Vizza et al., 2018) and Torin (Wang et al., 2015) and autophagy inhibitors such as 3-MA (Wang, Li & Chen, 2018) and Bafilomycin A1 (Wang et al., 2017) are conventional drug induction methods. As a classical mTOR inhibitor, RAPA can effectively induce stem cells to increase their autophagy levels in a dose-dependent manner (Sotthibundhu et al., 2016). RAPA induces autophagy by binding the mammalian RAPA target protein (mammalian target of RAPA, mTOR), causing mTOR signaling pathway cascade molecules to become active such that cells translate autophagy-related proteins (Wang & Zhang, 2019). The dynamic observation of the number and morphology of autophagosomes under electron microscopy is the gold standard for autophagy detection. The number of autophagosomes in cells represents the level of autophagy to some extent (Soto-Burgos et al., 2018). In this study, we observed the number and morphology of autophagosomes in cells by TEM. MDC dye is eosinophilic fluorescent staining that is a specific marker stain for detecting autophagosome formation (Gan et al., 2017). In this study, we observed autophagy levels qualitatively by microscope and quantitatively by FCM.

Multiple factors regulate mesenchymal stem cell fate decisions. For example, the differentiation of MSCs into osteoblasts depends on physical, chemical, and biological factors (Rodolfo, Bartolomeo & Cecconi, 2016). The level of autophagy decreases when cells differentiate into bone-like cells. The inhibition of autophagy could affect the survival and differentiation of MSCs (Gómez-Puerto et al., 2016). By detecting changes in autophagic flow and expression of osteogenic markers, autophagy is implicated in the osteogenic differentiation of MSCs. The osteogenic potential reduces when autophagy activity increases in MSCs. Following pretreatment with the autophagy inhibitor 3-MA and the autophagic agonist RAPA, BMMSCs were transplanted in nude mice, and in vivo analysis showed that the ectopic osteogenesis quality increased and that the bone-like tissue increased in the RAPA-treated group; the recovery of the 3-MA-treated osteogenesis ability was inhibited (Wan et al., 2017). Previous studies showed that RAPA promoted the osteoblastic differentiation of human embryonic stem cells by blocking the PI3K/AKT/mTOR pathway and stimulating the BMP/Smad pathway (Arianna et al., 2017; Lee et al., 2010). Also, there are recent reports of studies on the epigenetics of MSMSC osteogenic differentiation. Among them, lnc-NTF3-5 and hsa_circRNA_33287 can positively regulate osteogenic differentiation of MSMSCs (Peng et al., 2018; Peng et al., 2019), and miR-1827 can negatively regulate osteogenic differentiation of MSMSCs (Zhu et al., 2017). It seems that further study may be related to epigenetic regulation of autophagy on the osteogenic differentiation of MSMSCs.

A previous study showed that RAPA did not have an osteogenic effect on MSCs but inhibited osteogenic differentiation induced by dexamethasone (Isomoto et al., 2007). However, the concentration of RAPA in this paper was 0.01 nM. RAPA inhibited endogenous mTOR with IC50 values of ∼0.1 nM in HEK293 cells (Edwards & Wandless, 2007). Nevertheless, it does not mean that the effective concentration of RAPA for osteogenic differentiation is in such a low nanomolar range.

Overtime in the osteogenic culture, the difference in Becn1 and LC3 expression between the four groups became less. It indicated that the effect of RAPA on MSMSCs gradually decreased over time, and the autophagic level decreased with differentiation. In this study, we used a variety of osteogenic markers to detect osteogenic differentiation. However, the autophagic marker was insufficient to illustrate the inherent relationship between autophagy and osteogenic differentiation. Thus, in future studies, more autophagic markers would be adopted. Moreover, to identify the effect, methods of gene silencing and over-expression would be used.

A few limitations should also be noted in this study. First, we did not perform a dose–response assay, which led to a lack of justification for the concentrations of RAPA. Second, we did not provide a MSMSCs proliferation assay result after four concentrations of RAPA treatment. The potential cytostatic effects of RAPA might influence the osteogenic differentiation of MSMSCs. Third, we did not select good autophagic markers and tested them on the 7th, 14th, and 21st day. The evidence above could not link the RAPA-mediated induction of autophagy with the pro-osteogenic response. Fourth, we did not semi-quantify the results of the in vivo ectopic bone formation. Some nude mice died in the duration of transplantation, which caused an insufficient sample size of biological repetitions.

According to this study, RAPA seems to be the right choice for the atrophic posterior maxilla. We might suggest adding an appropriate amount of RAPA to bone substitutes when doing maxillary sinus membrane elevation. Otherwise, RAPA modified bone substitutes may show a better ability to osteogenesis. Furthermore, some older adults may prevent atrophic posterior maxilla by taking oral medications of RAPA.

Conclusions

In conclusion, the results indicated that RAPA promoted the osteogenic differentiation of MSMSCs within a specific range. Autophagy might play an essential role in the osteogenic differentiation of MSMSCs. The findings of the study also provided new insight into the prevention and treatment of atrophic posterior maxilla.

Supplemental Information

Supplemental Information 1 Raw data of qPCR

Click here for additional data file.

Thanks for rats and mice involved in the experiments and the 900th Hospital of Joint Logistics Support Force platform of the Chinese People’s Liberation Army. We express our gratitude to anonymous reviewers for providing insightful comments.

Additional Information and Declarations

Competing Interests

Author Contributions

Animal Ethics

Data Availability

The authors declare there are no competing interests.

Yanjun Lin conceived and designed the experiments, performed the experiments, analyzed the data, prepared figures and/or tables, authored or reviewed drafts of the paper, and approved the final draft.

Min Zhang performed the experiments, prepared figures and/or tables, and approved the final draft.

Lin Zhou analyzed the data, prepared figures and/or tables, and approved the final draft.

Xuxi Chen analyzed the data, authored or reviewed drafts of the paper, and approved the final draft.

Jiang Chen and Dong Wu conceived and designed the experiments, authored or reviewed drafts of the paper, and approved the final draft.

The following information was supplied relating to ethical approvals (i.e., approving body and any reference numbers):

Institutional Ethics Committee of the School of Stomatology, Fujian Medical University provided full approval for this research (Ref. [2017] NO.59).

The following information was supplied regarding data availability:

The raw data of qPCR are available in the Supplementary File.

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
