# Peer review of "Promoting effect of rapamycin on osteogenic differentiation of maxillary sinus membrane stem cells"

_PeerJ, doi:10.7717/peerj.11513_

## Round 0.1 · original submission · Major Revisions

Please be sure to fully address both reviewers' comments.

Reviewer 1 ·

Basic reporting

Clear and unambiguous, professional English used throughout
Introduction needs more details.
- In the literature there is a paper by Isomoto et al: "Rapamycin as an inhibitor of osteogenic differentiation in bone marrow-derived mesenchymal stem cells" published in 2007 in the Journal of Orthopedic Science, I suggest the authors should better describe the different studies that demonstrate how rapamycin induces osteogenic differentiation in stem cells.
- Rows 69-70; this paragraph should be reworded as several papers report the role between autophagy and osteogenic differentiation of BMSCs even if not MSMSCs (see Pantovic A, Krstic A, Janjetovic K, et al. Coordinated time-dependent modulation of AMPK / Akt / mTOR signaling and autophagy controls osteogenic differentiation of human mesenchymal stem cells. Bone. 2013; 52 (1): 524-531. doi: 10.1016 / j.bone.2012.10.024, Vidoni C, Ferraresi A, Secomandi E, et al Autophagy drives osteogenic differentiation of human gingival mesenchymal stem cells. Cell Commun Signal. 2019; 17 (1): 98. Published 2019 Aug 19. doi: 10.1186 / s12964-019-0414-7; Wan Y, Zhuo N, Li Y , Zhao W, Jiang D. Autophagy promotes osteogenic differentiation of human bone marrow mesenchymal stem cell derived from osteoporotic vertebrae. Biochem Biophys Res Commun. 2017; 488 (1): 46-52. Doi: 10.1016 / j.bbrc.2017.05.004 )
- Since the authors carry out their ex vivo experiments with biosynthetic scaffolds, I would also suggest describing how the combined presence of synthetic scaffolds (Bio-Oss) and rapamycin promotes osteogenic differentiation (see Lee KW et al. Rapamycin promotes the osteoblastic differentiation of human embryonic stem cells by blocking the mTOR pathway and stimulating the BMP / Smad pathway. Stem Cells Dev 2010; 19: 557–568. Carpentieri A. et al. "Rapid Rapamycin-Only Induced Osteogenic Differentiation of Blood-Derived Stem Cells and Their Adhesion to Natural and Artificial Scaffolds. "Stem cells international vol. 2017 (2017): 2976541. doi: 10.1155 / 2017/2976541).
- In Discussion, rows 271 and 320 the role of Runx2, as well as epigenetic changes, during osteogenic differentiation induced by rapamycin and scaffold have recently been described. Literature should be added (see Gambacurta A. et al. Human osteogenic differentiation in Space: proteomic and epigenetic clues to better understand osteoporosis. Sci Rep. 2019; 9 (1): 8343. Published 2019 Jun 6. doi: 10.1038 / s41598-019 -44593-6).


- Structure conforms to PeerJ standards
- Figures are good quality, well labelled and described.
- Raw data are supplied

Experimental design

- Original primary research within Scope of the journal.
- Research question well defined, relevant & meaningful.
- Rigorous investigation performed to a good technical & ethical standard.
- Methods described with sufficient detail & information to replicate.

Validity of the findings

- The impact and novelty are evaluated positively
- All underlying data have been provided; they are robust, statistically sound, & controlled.

Additional comments

The paper entitled "Promoting effect of rapamycin on osteogenic differentiation of maxillary sinus membrane stem cells" by Yan-Jun Lin and co-authors describe how rapamycin is able to induce stem cells osteogenic differentiation (in this study maxillary sinus membrane stem cells) and suggest a relationship between osteogenic differentiation and autophagy. The design of this study shows sufficient data in order to confirm the potential role of rapamycin in osteogenic differentiation and how there is a strong link with the autophagy process.
In conclusion, I feel that this paper, after revision, is of a high enough scientific bases to be considered for publication in PeerJ.

·

Basic reporting

Manuscript is well written and clearly articulated.

Background is mostly appropriate, however, there is a lack of detail as to the effects of rapamycin on other MSC populations and genetic models where the role of mTORC1 in lineage determination and osteogenesis has been examined in detail.

Structure conforms to PeerJ standards

Figures are ok and legends mostly ok.

Experimental design

Research question:
The effects of rapamycin and mTORC1’s role in osteogenesis had been examined in detail in human and rodent cells and rodent models. In this manuscript, the authors use rat MSMSCs as a model system to examine the effects of rapamycin on osteogenesis. More detailed studies as to the differentiation potential of human MSMSCs has been performed previously but not using rat cells.
The link to autophagy is tenuous at best and at no point do the authors test the hypothesis that rapamycin promotes osteogenesis in rat MSMSCs by suppressing mTORC1-dependent autophagy. For the osteogenic assays, the experiments haven’t been performed where the cytostatic effects of rapamycin have been accounted for and there is no justification for the rapamycin concentrations used in the study.

Validity of the findings

As pointed out herein, there are a number of limitations to the experiments conducted that should be addressed by the authors:

Can the authors justify the use of a single dose of rapamycin? Rapamycin/FBK12 has high affinity for mTORC1 and may be considered irreversible which could be used to justify a single dose. However, the authors provide no supportive data to this effect (test activation status of p70S6k and/or 4E-BP1 +/- Rapamycin time course). A single dose may be insufficient to inhibit mTORC1 for the duration of the assay which could affect the interpretations of the study findings.

In terms of the dose of rapamycin, these are not justified. Rapamycin has an IC50 for inhibition of mTORC1 in the low nanomolar range (<1nM). Therefore, in terms of the experimental design, the authors should show the concentrations of Rapamycin that inhibit mTORC1 is rat MSMSCs by performing a dose response and western blotting for the activation status of mTORC1 substrates.

The authors have quantified alizarin red staining as a readout of osteogenesis/mineralisation. While this method does provide some indication, it is not considered to be an accurate method and the authors would be better placed to measure acid solubilised mineral formation and normalise to total cell number.

With respect to the Alkaline phosphatase and alizarin red quantitation, the assays do not consider the effects of Rapamycin on cell proliferation. Rapamycin is cytostatic and thus the signal for the ALK P and ARS must be normalised to cell number as the differences in the staining could be attributed to changes in cell number.

With respect to the ectopic bone assay, the authors provide no quantitation (histomorphometry using Image J or Osteomeasure) of the explants. Thus, comments such as “rapamycin markedly induced bone-like tissue” are very subjective and the authors should perform a more detailed analysis of the explants which may provide some insight into how rapamycin may function to promote osteogenesis in rat MSMSCs (i.e. is there more osteoblasts? If not, then an increase in OB function?).

In terms of induction of autophapy, the authors provide TEM analysis showing putative autophagosomes and qRT-PCR data showing induction of Benc1 and LC3 expression. In terms of the autophagy field, these two measures are deemed insufficient to conclude induction of autophagy and the authors would need to provide, at the very least, LC3 I/II protein levels in response to rapamycin.

The authors have suggested that Rapamycin promotes autophagy which is a mechanism by which rapamycin promotes oteogenesis. However, as stated by the authors in the final discussion, no experiments were performed to test this hypothesis. Moreover, there is no correlation between the level of induction of autophagy (Fig. 2C; highest induction at 1uM) and osteogenic induction (Fig. 3C; 1uM has poor effect).

In terms of experimental details, these appear to be ok. The authors should make sure that the full name of the chemical (for example MDC (Monodansylcadaverine) is defined.

In terms of basic characterisation of rat MSMSCs, it would be of interest to enumerate the number of CFU-F’s in this tissue which would be a good addition to Figure 1.

---

## Round 0.2 · Major Revisions

Please address the remaining issues of the reviewer and revise your manuscript accordingly.

·

Basic reporting

no comment

Experimental design

no comment

Validity of the findings

no comment

Additional comments

My 3 major issues with the initial version of the manuscript was a lack of justification for the concentrations of Rapamycin used, the potential cytostatic effects of rapamycin and the lack of evidence linking the rapamycin-mediated induction of autophagy with the pro-osteogenic response.

The authors have decided not to provide any dose-responses and instead quote other MSC literature, which is in a different cell context. Given that this is the first study using rat MSMSCs, a rapamycin dose-response seems appropriate.

With respect to the cytostatic effect of rapamycin, the cytostatic effects may be limited given that the cells are only treated for 4 hours and the differentiation assays go for several weeks. Nonetheless, as a basic measurement of the cellular response to rapamycin, a short proliferation assay (3-4 days) would provide useful information about how this cell population responses to drug treatment (given the potential clinical applications).

With respect to the pro-osteogenic response of rapamycin being related to autophagy, the authors have revised the discussion to make it clearer that no conclusions can be drawn from their data to suggest this concept. As I noted previously, the pro-osteogenic response and stimulation of autophagy do not appear to correlate (i.e. at the dose of rapamycin that stimulates maximal autophagy (1mM), the osteogenic response is diminished).

With respect to quantitation of the in vivo ectopic bone formation, if the authors fail to provide any quantification of this data, then they must refrain from using any language that suggests the data has been quantified ie “more significant” (line 262) and ‘markedly induced” (line 258). Moreover, ImageJ (Fiji plugin) and Osteomeasure are widely used for histomorphometry and the authors shouldn’t dismiss these approaches. The impact of their in vivo model would be more compelling if new bone area or osteocycte numbers or osteocyte numbers/bone area were measured.

Overall, the authors could mitigate some of these issues by expanding the section in the discussion where they describe the current limitations of the study.

---

## Round 0.3 · accepted · Accept

All remaining issues were addressed and the revised manuscript is acceptable now.

·

Basic reporting

no comment

Experimental design

no comment

Validity of the findings

no comment

Additional comments

The manuscript describes, for the first time, the effects of rapamycin on the osteogenic potential of a unique stem cell population, maxillary sinus membrane stem cells. In the revised manuscript, the authors have detailed the limitations of their study such that a reader will be sufficiently informed as to the limitations of the data.